# Umbrella Review: Stress Levels, Sources of Stress, and Coping Mechanisms among Student Nurses

Leodoro J. Labrague 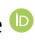

Marcella Neihoff School of Nursing, Loyola University Chicago, Chicago, IL 60611, USA; leo7_ci@yahoo.com

**Abstract:** Prelicensure nursing students face significant stress from their education and clinical placements, highlighting the crucial need for the development of effective coping mechanisms with which to manage both academic and clinical responsibilities, ultimately enhancing the wellbeing and academic performance of these students. This umbrella review aims to evaluate and synthesize existing review articles that examine stress levels and coping mechanisms among student nurses during their education and training. Five databases (PsycINFO, PubMed, CINAHL, Scopus and Web of Science) were searched for review articles published from 2010 onwards. This review includes twelve articles, encompassing 189 studies. The review findings demonstrate that student nurses experience moderate-to-high levels of stress during their nurse education. Major sources of stress include academic demands, patient care responsibilities, and interactions with nursing staff and faculty. Commonly utilized coping skills involve problem-solving behaviors, transference, and maintaining an optimistic outlook. Given the adverse consequences of stress, nurse educators play a critical role in the development of strategies with which to reduce stress and enhance coping skills among student nurses. This study was not registered.

**Keywords:** stress; coping; nursing; education; students; integrative review; systematic review

## 1. Introduction

Stress and coping during nurse education and training are widely recognized as important areas of research, as nursing students often experience high levels of stress due to academic demands, clinical placements, and personal life stressors [1,2]. Stress, universally defined, is a physiological and psychological response to a perceived threat or challenge. It involves the body's adaptive reactions aimed at mobilizing resources to cope with the demands of a situation. Stress can manifest as a complex interplay of physical, emotional, and behavioral changes, and its intensity varies based on individual perceptions and coping mechanisms [3,4]. Although stress in general is often considered harmful, when maintained at manageable levels, it can potentially offer benefits by serving as a motivational force for students, fostering resilience, and encouraging the development of effective coping strategies [5].

Numerous studies have identified common stressors in nursing students, including heavy academic workloads, time pressures, clinical placements, and personal life challenges such as financial problems and family issues [4,6]. Prolonged exposure to excessively high stress levels can have detrimental effects on the psychological health, academic performance, and overall wellbeing of student nurses [5,6]. Therefore, effective coping strategies are essential for reducing stress, promoting wellbeing, and fostering academic success among nursing students.

When faced with stress, nursing students commonly employ a combination of problem-centered coping strategies and emotion-centered coping mechanisms [7,8]. Problem-centered coping mechanisms involve seeking social support, managing time effectively, and engaging in active problem-solving. These strategies target the root causes of stress and provide long-term stress relief [8,9]. On the other hand, emotion-focused coping strategies

encompass positive reinterpretation, acceptance, and mindfulness. While these strategies may help mitigate and manage the behavioral responses to stress, they offer only short-term stress reduction as they do not directly address the underlying causes of stress [9,10].

This umbrella review is guided by Lazarus and Folkman's transactional model of stress and coping [9]. This influential framework emphasizes that individuals engage in coping strategies based on their appraisal of stressors, employing both problem-centered and emotion-centered mechanisms [9]. In the context of nursing students, the model provides a comprehensive lens through which to understand how they navigate stress, addressing not only the identification of stress levels and sources but also shedding light on the effectiveness of the coping mechanisms employed. By utilizing this theoretical foundation, the research aims to explore the dynamic interplay between stressors and coping strategies, offering insights applicable to the development of targeted interventions in nursing education and practice.

Over the past three decades, stress and coping in student nurses have been extensively researched, resulting in a substantial volume of individual studies and literature reviews [11,12]. Guided by the Joanna Briggs Institute (JBI) methodology, this umbrella review aims to examine the current state of knowledge on stress and coping in student nurses during their prelicensure programs [13,14].

In the context of nurse education and training, the results of this umbrella review are crucial for several reasons in the context of nursing practice for students. Firstly, understanding the specific stressors encountered by student nurses and the sources of their stress is essential for the development of targeted interventions by which to promote mental wellbeing during their education. By comprehensively synthesizing existing literature, this review aims to identify common stressors, such as academic pressures, clinical demands, and interpersonal challenges, and thus provide valuable insights for educators and institutions to tailor support mechanisms. Moreover, a detailed examination of coping mechanisms utilized by student nurses is vital for informing evidence-based strategies that can be incorporated into nursing education programs. Identifying effective coping strategies is not only beneficial for students' mental health but also crucial for enhancing their resilience and their ability to manage stressors in their future nursing careers. This review aims to contribute to the ongoing efforts in nursing education to create environments that foster student wellbeing, reduce burnout, and ultimately cultivate a workforce that is better equipped to provide high-quality and compassionate patient care. The findings from this umbrella review have the potential to guide policy decisions, curriculum development, and support services, ultimately shaping the landscape of nursing education and practice. Despite the abundance of literature reviews examining stress and coping in students, no umbrella review synthesizing findings from previous reviews has been found to date.

The aim of this research was to conduct an umbrella review to systematically synthesize and analyze existing literature on stress levels, sources of stress, and coping mechanisms among student nurses. This overarching review seeks to provide a comprehensive understanding of the multifaceted aspects of stress experienced by student nurses during their education. Specifically, the research aimed to identify common stressors, explore variations in stress levels across different educational settings, and critically evaluate the effectiveness of coping mechanisms employed by students.

## 2. Materials and Methods

### 2.1. Design

An umbrella review is a systematic review that summarizes and evaluates the findings of multiple systematic reviews and research syntheses on a specific topic [15]. An umbrella review synthesizes and evaluates the quality, quantity, and consistency of evidence from multiple reviews, providing a broader and more reliable understanding of a specific research question or topic [13]. By integrating and summarizing the findings from various reviews, an umbrella review offers a more comprehensive and nuanced perspective on a particular field of study [16]. This type of review can be particularly valuable when guiding

clinical practice and decision-making, informing clinical guidelines, public health policies, and future research directions [15,17].

Belbasis et al. [15] have provided a guideline for conducting an umbrella review. Their guideline includes steps such as clearly defining the research question, establishing inclusion and exclusion criteria, conducting a literature search, extracting, and analyzing data, evaluating the strength of evidence, and summarizing and presenting the data. Their result was presented using the Preferred Reporting Items for Systematic Reviews and Meta-Analyses (PRISMA).

## 2.2. Sources of Data and Search Strategy

To locate relevant literature reviews, five databases (SCOPUS, Web of Science, PubMed, CINAHL and PsychINFO) were searched using the following search terms: "student nurses" OR "prelicensure student nurses" AND "stress" OR "psychological distress" AND "coping mechanisms" OR "coping skills" AND "nurse education" OR "clinical practice" AND "integrative review" OR "systematic review" OR "scoping review" OR "literature review". The inclusion criteria for articles were as follows: (a) peer-reviewed quantitative reviews of original studies assessing stress and coping among prelicensure nursing students, (b) published in the English language and (c) published from 2010 onwards. For the purpose of homogeneity, this review was limited to quantitative reviews. Focusing exclusively on quantitative studies in this umbrella review ensures a rigorous and systematic analysis of objective, numerical data related to stress levels, sources of stress, and coping mechanisms among student nurses. This approach allows for a standardized synthesis of evidence, enabling the identification of patterns, trends, and statistically significant associations that contribute to a more robust and generalizable understanding of the topic.

## 2.3. Search Outcomes

The initial search identified a total of 189 articles examining stress and coping in student nurses. These studies were then screened and filtered for duplicates, resulting in the removal of 91 reviews. The remaining 98 articles were further evaluated based on the inclusion criteria, leading to the exclusion of an additional 71 articles. The full texts of the remaining 27 articles were read, and 12 articles were found to be relevant for the review (Figure 1).

## 2.4. Quality Appraisal

The JBI Critical Appraisal Checklist for Systematic Review and Research Syntheses was used to assess the quality of the gathered evidence. This checklist consists of 11 items that can be answered with yes (1), no (0) or unclear (0), with a maximum score of 11. The quality of each review was categorized as low (<5.5), moderate (5.5–8), or high (9–11).

## 2.5. Data Extraction and Synthesis

Data extraction and synthesis were conducted by two researchers. The primary researcher, who was also the author of the study, worked alongside a second independent researcher not affiliated with the study to enhance objectivity and reliability in the process. This dual-researcher approach aimed to enhance the robustness and validity of the data extraction and synthesis process, contributing to the comprehensive analysis of stress levels, sources, and coping mechanisms among student nurses. To facilitate result comparison, a matrix table was created, and the following information was extracted: author, country, review type, databases used, number of studies included, key findings, and the quality appraisal checklist employed. The results of each review were synthesized based on the formulated research questions.

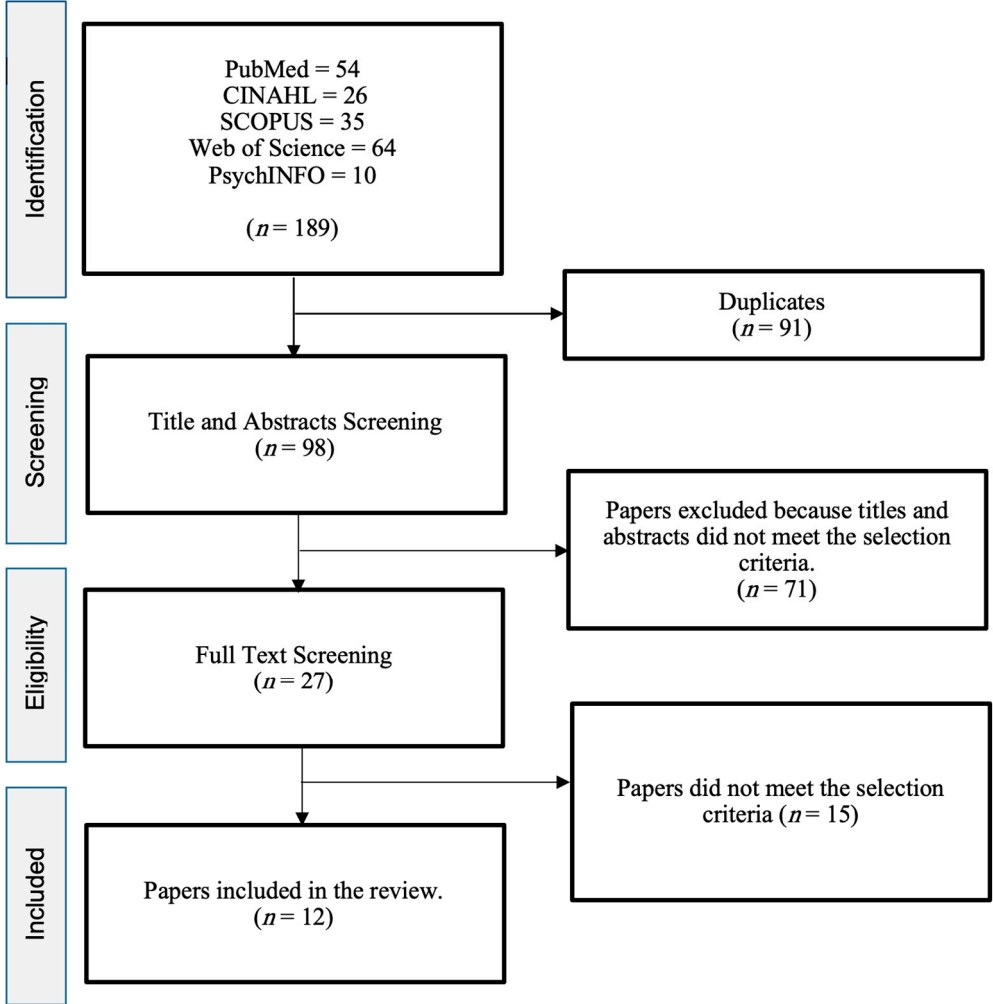

**Figure 1.** Diagram of the process used to identify references for the review.

## 3. Results

### 3.1. Characteristics of the Included Review Articles

Twelve reviews were included in this umbrella review. Among these, six utilized an integrative review design [3,8,18–21], three employed a systematic review design [5,22,23], two used a scoping review design [24,25], and one conducted a systematic review with a meta-analysis [11]. The number of studies included in each review ranged from 8 to 27, resulting in a total of 189 studies (Table 1). The most commonly used databases for retrieving relevant articles were CINAHL, MEDLINE, Scopus and PubMed. Seven studies reported the quality appraisal checklist used to assess the methodological rigor of the reviews, with the Critical Appraisal Skills Programme (CASP) being employed in three of them. Among the 12 reviews, 2 were deemed to have moderate quality, while the remaining 10 were rated as high quality (Table 2).

**Table 1.** Summary of studies included.

| Author | Country | Review Type | Databases Used | Number of Studies | Key Findings | Quality Appraisal Checklist |
|---|---|---|---|---|---|---|
| Alatawi et al. [24] | Saudi Arabia | Scoping review | PubMed, CINAHL, EBSCO, ProQuest and Medline | 22 | • Stress level: moderate to high.<br>• Sources of stress: academic stressors (academic loads, time management, fear of examination), clinical stressors (nursing staff expectations, caring for patients), and personal stressors (financial distress, death of family). | NR |
| Alzayyat et al. [18] | Jordan | Integrative review | MEDLINE, CINAHL, PsycINFO and PubMed | 13 | • Sources of stress: academic demands, relations in the clinical environment and caring for patients and families.<br>• Academic year level was not associated with stress levels. | NR |
| Bhurtun et al. [21] | Finland | Integrative review | MEDLINE, CINAHL, PsycINFO and PubMed | 13 | • Stress level: moderate to high levels of stress during their clinical training.<br>• Sources of stress: teachers and nursing staff were a strong stressor.<br>• Top coping skills: problem solving and transference the most common coping techniques. | NR |
| Labrague et al. [3] | Oman | Integrative review | PsycINFO, PubMed, CINAHL, MEDLINE and Scopus | 27 | • Top coping skills: problem focuses coping strategies—problem solving behaviors, self-confident behaviors.<br>• Least frequently used coping skills: avoidance. | QualSysts |
| Labrague et al. [8] | Oman | Integrative review | CINAHL, MEDLINE, PsycINFO and PubMed | 13 | • Stress level: moderate levels of stress.<br>• Sources of stress: caring for patients, assignments and workloads, and negative interactions with staff and faculty.<br>• Top coping skills: problem-solving strategies such as developing objectives to resolve problems, adopting various strategies to solve problems, and finding the meaning of stressful events. | Critical Appraisal Checklist |

**Table 1.** *Cont.*

| Author | Country | Review Type | Databases Used | Number of Studies | Key Findings | Quality Appraisal Checklist |
|---|---|---|---|---|---|---|
| Labrague et al. [22] | Oman | Systematic review | SCOPUS, CINAHL, PubMed and Ovid | 11 | • Stress level: moderate-to-high stress levels.<br>• Sources of stress: heavy workloads, taking care of patients.<br>• Top coping skills: problem solving behaviors.<br>• Least frequently used coping skills: avoidance.<br>• Higher level students experiencing higher stress levels. | Critical Appraisal Checklist |
| Majrashi et al. [25] | Saudi Arabia | Scoping review | CINAHL, MEDLINE and PubMed, | 13 | • Sources of stress: stress from distance learning, stress from assignment and workloads, clinical training, stress from COVID-19 infection.<br>• Top coping skills: seeking information, staying optimistic, transference. | Hawker's Quality Assessment Tool |
| McCarthy et al. [19] | Ireland | Integrative review | CINAHL, PubMed and PsycINFO | 25 | • Sources of stress: caring for patients, relationship with clinical colleagues and faculty, academic environment, examinations, and assignments.<br>• Top coping skills: problem solving, staying optimistic, transference, and social support. | Critical Appraisal Tool |
| Pulido-Martos et al. [23] | Spain | Systematic review | MEDLINE and PsycInfo | 23 | • Sources of stress: academics (reviews, workload and problems associated with studying, among others).<br>• Other sources of stress include clinical sources (such as fear of unknown situations, mistakes with patients or handling of technical equipment).<br>• No changes occur at the different years of the student's education. | NR |
| Younas [20] | Canada | Integrative review | PubMed, EMBASE, Cochrane, CINHAL, ASSIA, PsycInfo, Science Direct and Google Scholar | 9 | • Stress level: moderate stress levels.<br>• Sources of stress: assignment workloads, lack of clinical knowledge, inadequate training, long clinical hours.<br>• Top coping skills: problem solving behaviors, transference, optimism, seeking family, and professional support. | NR |

Table 1. *Cont.*

| Author | Country | Review Type | Databases Used | Number of Studies | Key Findings | Quality Appraisal Checklist |
|---|---|---|---|---|---|---|
| Zheng et al. [11] | China | Systematic Review with meta-analysis | PubMed, Cochrane, Web of Science, CNKI and China Biomedical Literature Service System | 8 | • The prevalence of stress among practicing nursing students was estimated to be 61.97%, suggesting a high prevalence of psychological stress among practicing nursing students. | Agency for Healthcare Quality and Research (AHRQ) |
| Li & Hasson [5] | Ireland | Systematic Review | CINAHL, Web of Science, Medline (OVID), PsycInfo, CNKI, WanFang Data, VIP and CMB | 12 | • Stress level: moderate-to-high stress levels.<br>• Interaction between resilience and stress and wellbeing was high.<br>• Resilience and low stress were found to better predict wellbeing. | Critical Appraisal Skills Programme (CASP) |

NR = not reported.

Table 2. JBI Critical Appraisal Checklist Criteria for Systematic Review and Research Syntheses.

| Reference | 1 | 2 | 3 | 4 | 5 | 6 | 7 | 8 | 9 | 10 | 11 | Total |
|---|---|---|---|---|---|---|---|---|---|---|---|---|
| Alatawi et al. [24] | 1 | 1 | 1 | 1 | 1 | 0 | 1 | 1 | 0 | 1 | 1 | 9 |
| Alzayyat et al. [18] | 1 | 1 | 1 | 1 | 1 | 0 | 1 | 1 | 0 | 1 | 1 | 9 |
| Bhurtun et al. [21] | 1 | 1 | 1 | 1 | 1 | 1 | 1 | 1 | 0 | 1 | 1 | 10 |
| Labrague et al. [3] | 1 | 1 | 1 | 1 | 1 | 1 | 0 | 1 | 0 | 1 | 1 | 9 |
| Labrague et al. [8] | 1 | 1 | 1 | 1 | 1 | 1 | 1 | 1 | 0 | 1 | 1 | 10 |
| Labrague et al. [22] | 1 | 1 | 1 | 1 | 1 | 1 | 1 | 1 | 0 | 1 | 1 | 10 |
| Majrashi et al. [25] | 1 | 1 | 1 | 0 | 1 | 1 | 1 | 1 | 0 | 1 | 1 | 9 |
| McCarthy et al. [19] | 1 | 1 | 1 | 0 | 1 | 1 | 1 | 1 | 0 | 1 | 1 | 9 |
| Pulido-Martos et al. [23] | 1 | 1 | 1 | 0 | 1 | 0 | 1 | 1 | 0 | 1 | 1 | 8 |
| Younas [20] | 1 | 1 | 1 | 1 | 1 | 0 | 0 | 1 | 0 | 1 | 1 | 8 |
| Zheng et al. [11] | 1 | 1 | 1 | 1 | 1 | 1 | 1 | 1 | 1 | 1 | 1 | 11 |
| Li & Hasson [5] | 1 | 1 | 1 | 1 | 1 | 1 | 1 | 1 | 0 | 1 | 1 | 10 |

Note: yes = 1; no/unclear = 0; 1 = is the review question clearly and explicitly stated?; 2 = were the inclusion criteria appropriate for the review question?; 3 = was the search strategy appropriate?; 4 = were the sources and resources used to search for studies adequate?; 5 = were the criteria for appraising studies appropriate?; 6 = was critical appraisal conducted by two or more reviewers independently?; 7 = were there methods to minimize errors in data extraction?; 8 = were the methods used to combine studies appropriate?; 9 = was the likelihood of publication bias assessed?; 10 = were recommendations for policy and/or practice supported by the reported data?; 11 = were the specific directives for new research appropriate?

### *3.2. Major Findings*

The results of the data synthesis were categorized into four themes: (a) moderate-to-severe stress levels, (b) sources of stress, (c) frequently used coping skills, and (d) stress in relation to academic levels.

### 3.2.1. Moderate-to-Severe Stress Levels

Seven reviews reported stress levels in student nurses ranging from moderate [3,20] to severe [5,11,21,22,24]. Three of these reviews utilized an integrative design, two employed a systematic design, one used a scoping review, and one conducted a systematic review

with a meta-analysis. All reviews analyzed articles on stress among students from various countries, except for the work of Labrague et al. [22], which focused specifically on Saudi Arabian student nurses, and the work of Younas [20], which included studies conducted only in Asia. Qualitative synthesis of the review results was conducted in all reviews except for that of Zheng et al. [11], who performed a meta-analysis to estimate the prevalence of psychological stress in practicing student nurses. The meta-analysis revealed a prevalence rate of 61.97%, indicating a higher prevalence of psychological stress among pre-licensure nursing students.

### 3.2.2. Sources of Stress

Under this theme, stress in the context of nursing education is defined as the psychological and emotional strain [9] experienced by students due to the demanding nature of academic coursework, clinical training, and interpersonal dynamics within educational settings, potentially impacting their overall wellbeing and academic performance. The top sources of stress identified in student nurses during nurse education and training were 'academic demands', 'caring for patients' and 'interaction with nursing staff and faculty'.

### Academic Stress

This subtheme pertains to the psychological strain experienced by nursing students in response to the demanding nature of their coursework, examinations, and high academic expectations, collectively referred to as academic stress [9]. Seven reviews identified 'academic demands' as the primary source of stress for student nurses during their education and training. Three integrative reviews identified heavy academic workloads [3,18,19] as the main stressor, while two systematic reviews [23,24] reported unreasonable assignments, tests, and examinations as major stressors for students. Similar findings were observed in reviews assessing stress and coping among Saudi Arabian student nurses [22] and in select Asian countries [20]. A scoping review examining stress and coping during the height of the pandemic identified academic demands and virtual learning as major stressors [25].

### Stress from Caring for Patient

This subtheme encompasses the emotional and psychological strain experienced by nursing students, due to the challenges and responsibilities associated with providing care to individuals who are ill or in need of medical attention—referred to as stress from caring for patients [10]. Six reviews identified 'caring for patients' as the second most common source of stress for student nurses. Three integrative reviews identified caring for ill patients and dealing with families as the main sources of stress [8,18,19]. This finding was supported by a systematic review by Labrague et al. [22], in which Saudi Arabian students reported heightened psychological stress levels when caring for patients. Two scoping reviews, one conducted before the pandemic [24] and one during the pandemic crisis [25], examined clinical sources of stress among pre-licensure students. Both reviews identified stress associated with patient care as a common challenge during their clinical placements.

### Stress from Interaction with Staff and Faculty

Under this subtheme, stress from interactions with staff and faculty in the context of nursing education is characterized by the emotional and psychological strain experienced by students due to challenging or negative encounters, expectations, or dynamics with nursing faculty and staff members during their academic and clinical training [9]. 'Interaction with nursing staff and faculty' was reported as the third most common source of stress for students during their nursing program. This stressor was identified in three integrative reviews and one scoping review. In a scoping review of 13 studies, in addition to unreasonable academic workloads, many pre-licensure students reported elevated levels of stress when interacting with nursing staff and clinical instructors [24]. This finding was supported by three integrative reviews, where interaction with nursing staff and faculty was regarded as one of the top stressors for students [8,19,21].

### 3.2.3. Frequently Used Coping Skills

Coping skills in the context of nursing education refer to the adaptive strategies and mechanisms employed by students to effectively manage and navigate the various stressors, challenges, and demands inherent in their academic coursework, clinical training, and interactions with faculty and staff [10]. Three distinct coping strategies were identified: problem-solving behaviors, transference, and staying optimistic.

#### Problem Solving Behaviors

Under this subtheme, problem-solving behaviors, as a coping strategy in nursing education, refer to the deliberate and systematic approach employed by nursing students to identify, analyze, and resolve challenges or stressors encountered during their academic and clinical training [9]. Seven review articles reported 'problem-solving behaviors' as the top coping strategy employed by student nurses to deal with and manage their stressors. Three of these were integrative reviews, one was a systematic review, and the remaining three were scoping reviews. These reviews, which analyzed global studies, consistently identified problem-solving behaviors as the most frequently used coping skill among nursing students [3,8,21]. A systematic review focusing on stress and coping among nursing and midwifery students found that both groups were more inclined to use desirable coping skills, including problem-solving [19]. Another systematic review conducted during the pandemic also identified seeking information, a problem-solving skill, as the most frequently used coping mechanism [25]. Two of these reviews had a narrower scope, one specific to Saudi students [22], and the work of Younas [20] focusing on Asian nursing students. In the systematic review of 13 studies conducted in Saudi Arabia, problem-solving behaviors were identified as the most frequently used coping skills, while avoidance was the least frequently used [22]. Younas [20] also found problem-solving behaviors to be the top-rated coping skills among Asian nursing students based on the analysis and appraisal of nine studies.

#### Transference

Transference as a coping strategy in nursing education involves the unconscious redirection of feelings, attitudes, and expectations from past experiences or relationships onto current interactions with faculty, staff, or peers [10]. Four review articles identified 'transference' as the second most frequently used coping strategy among student nurses. Bhurtun et al. [21] appraised and synthesized 13 studies, identifying transference as the second most frequently used coping strategy after problem-solving behaviors. Younas [20], in a study analyzing stress and coping studies in an Asian context, found that student nurses used problem-solving behaviors and transference as coping skills. McCarthy et al. [19], analyzing 25 studies, found that both midwifery and nursing students utilized transference as a way of managing academic and clinical stress. A scoping review of 13 studies conducted during the pandemic [25] also identified transference as frequently used coping during the height of the pandemic crisis.

#### Staying Optimistic

Staying optimistic as a coping skill involves maintaining a positive and hopeful outlook, even in the face of challenges or stressful situations [9]. This includes adopting a mindset that focuses on positive possibilities, finding silver linings, and cultivating resilience to navigate difficulties with a hopeful attitude. Three review articles reported 'being optimistic' as the third most frequently used coping skill when confronted with stressors during clinical placements. Staying optimistic was identified as one of the main coping skills utilized by Asian students in an integrative review by Younas [20]. Similar results were obtained in a literature review of 25 studies that measured stress and coping in nursing and midwifery students [19]. In a study examining stress and coping among student nurses during the height of the COVID-19 crisis, Majrashi et al. [25] identified staying optimistic as one of the most frequently used coping skills, alongside seeking information.

3.2.4. Stress in Relation to Academic Year Level

Three review articles assessed how stress levels change according to academic year levels. Two reviews found no apparent changes in stress levels across academic year levels [18,23], while one review [22] found that academic year levels were associated with heightened stress levels in students. A systematic review of 23 studies found no apparent changes in stress levels as student nurses progressed to higher academic year levels [23]. Similarly, an integrative review by Alzayyat et al. [18] found no statistical difference in stress levels among student nurses across different year levels. In contrast, a systematic review of stress and coping studies in Saudi Arabia showed higher stress levels in student nurses in higher academic year levels compared with those in lower year levels [22].

## 4. Discussion

### 4.1. Key Findings

The literature has consistently identified stress as a common experience among health-related professions, with studies indicating a high prevalence of stress among this population. Earlier studies have reported that up to 80% of health profession students experience a high amount of stress during their academic years, which is supported by the findings of Liu et al. [1] and Ayaz-Alkaya and Simones [2]. Nursing students, in particular, face personal life challenges during their adolescent stage, and the nursing program itself exposes them to various stressors throughout their pre-licensure program [4,6]. Balancing academic studies with clinical rotations, which often involve long hours and unpredictable schedules, further contributes to the higher stress experience in student nurses. Across countries, the umbrella review reveals a relatively uniform intensity of stress, consistent sources of stress, and common coping mechanisms among student nurses. These findings underscore the global nature of challenges faced by nursing students during education and clinical training. In light of this, the implications for nursing education globally are significant, necessitating collaborative efforts to establish standardized support measures, culturally sensitive interventions, and the integration of flexible learning technologies with which to address the shared stressors experienced by nursing students worldwide.

This review identified the top stressors in student nurses, including academic demands, caring for patients, and interactions with nursing staff and faculty. Academic demands are expected to be the top stressor in nursing students due to the heavy workload of lectures, labs, clinical rotations, and assignments [5]. Clinical experiences, including caring for patients, can be emotionally and physically demanding, especially for students and new nurses. The responsibility of caring for sick and dying patients and the emotional toll of witnessing patients' suffering can overwhelm nursing students [26]. Interactions with nursing staff and faculty can also be a potential source of stress if students feel unsupported and lack guidance in their clinical training [5,21]. Nursing students may feel intimidated, particularly when faced with experienced staff nurses [27]. Student nurses may perceive interactions with faculty and nursing staff as stressful due to a variety of reasons. Firstly, a lack of clear communication and expectations can create uncertainty, leading to heightened anxiety among students [7]. Additionally, disparities in power dynamics, with students often perceiving faculty and staff as authority figures, may contribute to feelings of intimidation and hinder open communication [8,9]. Moreover, inadequate feedback and support in clinical settings can exacerbate stress, as students may struggle to navigate challenges without sufficient guidance [10]. Addressing these issues through clear communication, supportive mentorship, and constructive feedback can help alleviate stress and enhance the overall learning experience for student nurses. This result provides support to earlier research that has identified academic demands, caring for patients, and interactions with nursing staff and faculty as top sources of stress [9,18,20].

To effectively manage and deal with these stressors, individuals should utilize coping skills that provide long-term resolutions to stress, such as problem-solving behaviors rather than emotion-focused strategies [28]. This review identified three coping strategies frequently used by student nurses: problem-solving behaviors, staying optimistic, and

transference. Problem-solving behaviors and staying optimistic have been shown to be effective coping mechanisms in managing stress and challenging circumstances [9,10]. Optimism, which involves maintaining a positive outlook on life, helps student nurses remain motivated and resilient in difficult circumstances [10,28]. Problem-solving involves identifying issues, developing action plans, and implementing them, in turn enabling individuals to reduce stress and regain control [9]. These findings support prior reports that have identified problem-solving behaviors and optimism as important coping strategies for nursing and non-nursing students [4,6]. These findings coincide with earlier research, wherein student nurses identified these coping mechanisms as useful for dealing with and managing their stressors during nursing education and training [18,21,24]

In addition to problem-focused coping strategies, an emotion-focused strategy, including transference, was also identified as a coping mechanism used by students to deal with stressors. Transference is a psychological phenomenon that occurs when individuals unconsciously transfer feelings, attitudes, and expectations from one person or situation to another. Transference, however, is considered an undesirable coping mechanism as it can prevent individuals from developing healthy coping skills and addressing the root causes of stress [9,10]. Overreliance on transference can be counterproductive [25]. In nursing education, students often face demanding academic and clinical challenges that may be beyond their immediate control. Emotion-focused coping strategies become crucial as they navigate stressors like heavy workloads and high-pressure situations in healthcare settings. By teaching nursing students techniques such as mindfulness, self-care, and seeking peer support, educators can help them effectively manage their emotional responses, promoting resilience and wellbeing in the demanding context of nursing education.

Nurse faculty can promote problem-solving behaviors in nursing students by providing constructive feedback, encouragement, and modeling the use of problem-solving skills. By assisting students in identifying the root causes of their stress and brainstorming potential solutions, nurse faculty can help them effectively handle their stressors instead of projecting their emotions onto others or situations.

### 4.2. Limitations of the Study

A limitation of this study is that it exclusively focused on quantitative studies, thereby excluding valuable qualitative insights that could provide a deeper understanding of the subjective experiences related to stress, sources of stress, and coping mechanisms among nursing students. Additionally, the decision to restrict the review to studies published in the English language may introduce language bias, potentially omitting relevant findings from non-English literature that could offer diverse perspectives on the topic. Another limitation lies in the exclusion of articles published before 2010, which may overlook earlier research that could contribute historical context and highlight potential changes or trends in stress among nursing students over time. These limitations collectively underscore the need for future research to adopt a more inclusive approach, considering both qualitative and non-English literature, and exploring a broader timeframe to ensure a comprehensive understanding of stress in nursing education.

### 4.3. Future Research Directions

This umbrella review identified several critical points that can guide future literature reviews. Firstly, of the 12 reviews analyzed, 5 did not evaluate the methodological rigor of the included studies. Assessing the quality of evidence is crucial for determining the reliability and validity of research findings [29]. Without analyzing the quality of the studies, it becomes challenging to make informed decisions based on the review findings.

Meta-analysis is a powerful tool that can improve the reliability and quality of research findings, informing evidence-based practice and policy decisions [30]. In this umbrella review, only one review utilized meta-analysis to analyze the data, partly due to the heterogeneity of the included studies, such as variations in scales used. Future research synthesis

should consider statistical pooling or meta-analysis for a comprehensive evaluation of the evidence.

Although this umbrella review provides evidence regarding the prevalence of stress among nursing students across countries, the studies included often lack in-depth discussions considering specific contextual nuances related to each country. Consequently, due to this limited contextual analysis, comparing differences across countries that might affect stress and coping in students becomes challenging. Recognizing the influence of varying cultural, educational, and healthcare system factors on student experiences is crucial for a comprehensive understanding of stress in nursing education, emphasizing the need for future studies to adopt a more nuanced approach in exploring and reporting results within diverse international contexts. Therefore, to advance our understanding of the nuanced factors contributing to stress among nursing students, future studies should delve into the specific contextual nuances, considering the diverse educational, cultural, and healthcare landscapes across countries, to facilitate more meaningful cross-cultural comparisons and inform tailored interventions.

Furthermore, there is a need for reviews examining the interaction between stress and coping, specifically identifying which coping skills are most effective for nursing students in dealing with their stressors. Different coping strategies may be effective depending on the situation and individual characteristics and resources [9,10,28]. This knowledge is essential to assist nurse educators in identifying potential coping strategies that could effectively reduce stress in students. Overall, enhancing the methodological rigor of reviews, conducting meta-analyses when appropriate, and exploring the specific coping skills that best assist nursing students in managing their stressors will contribute to a deeper understanding of stress and coping in this population and inform evidence-based interventions and support strategies.

## 5. Relevance to Nurse Education

Stress during nurse education and clinical training can potentially exacerbate the current shortage experienced by healthcare institutions worldwide. Consequently, nurse educators play a crucial role in the development of stress reduction measures and enhancing coping skills in nursing students. Given that academic demands have been identified as the primary stressor among student nurses, it is vital for nurse educators to implement measures that assist students in effectively managing their workloads and reducing stress [31]. This may involve prioritizing essential coursework and assignments, increasing flexibility, and providing academic support [31]. By strengthening coping and social support, engaging in stress-reducing activities, and seeking professional help when needed, student nurses can effectively deal with stress related to patient care and improve their overall health and wellbeing [18,23]. The literature has identified several theoretically based interventions that are equally effective in reducing stress among students, including mindfulness-based interventions [32], behavioral-based stress management programs [33], and evidence-based resilience interventions [34].

To strengthen positive coping skills, nurse educators should focus on building and fostering problem-focused coping strategies in students to help them effectively deal with their stressors. Evidence has shown the importance of structured orientation programs for new students and structured faculty–student mentoring programs to assist students in developing active coping skills [35,36]. Social support, derived from family, relatives, and friends, should be strengthened as it has been found to be helpful in protecting students from the long-term effects of stress [36]. Additionally, nurse educators can model positive coping behaviors and share their own experiences of managing stress in a healthy way, leading to improved wellbeing and increased retention [37]. Collectively, these strategies can assist student nurses in bolstering their coping abilities and effectively managing the numerous stressors encountered during nurse education and training.

## 6. Conclusions

This umbrella review provides a new understanding of stress in nurse education by synthesizing evidence from multiple reviews and research syntheses. Globally, the intensity of stress and coping mechanisms among student nurses exhibits variation, yet the identified sources of stress remain remarkably consistent. While stress levels range from moderate to severe across different regions, the overarching themes of academic demands, patient care responsibilities, and interactions with nursing staff and faculty persist as primary stressors for students worldwide. Commonly used coping strategies included problem-solving behaviors, transference, and maintaining optimism. This review did not establish a relationship between academic levels and stress experience among pre-licensure nursing students. This suggests a universal need for targeted interventions and support strategies to address common stressors and enhance coping mechanisms among student nurses on a global scale.

**Funding:** This research received no external funding.

**Institutional Review Board Statement:** Not applicable.

**Informed Consent Statement:** Not applicable.

**Public Involvement Statement:** No public involvement in any aspect of this research.

**Guidelines and Standards Statement:** This manuscript was drafted against the Preferred Reporting Items for Systematic Reviews and Meta-Analyses.

**Conflicts of Interest:** The author declare no conflict of interest.

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
