# Peer review of "Umbrella Review: Stress Levels, Sources of Stress, and Coping Mechanisms among Student Nurses"

_nursrep, doi:10.3390/nursrep14010028_

Round 1

Reviewer 1 Report

Comments and Suggestions for Authors

This is an interesting manuscript with some findings and recommendations that can be implemented by nursing education. Corrections suggested: Explain and describe what is meant by "transference" as a way of coping as it seems to be not so effective and readers might not understand what it entails and why it is not so effective. Emotion-focused coping is discussed as not such an effective way of coping, but at times it can be an effective methods, eg. with issues that cannot be changed and need to be accepted, emotional regulation can be the most effective way of coping. Especially patient related problems cannot always be changed/addressed. The stress related to communication with faculty and clinical staff need to better discussed and also addressed in implications for nursing education - it seems to be a problem that needs attention from nursing education - one wonders if the faculty members are not approachable, or if the students lack communication skills - problems that can be surely be addressed. 

Minor corrections: See pdf document  

Comments on the Quality of English Language

Minor corrections

Author Response

Thank you for the constructive review. Below are my responses.

Corrections suggested: Explain and describe what is meant by "transference" as a way of coping as it seems to be not so effective and readers might not understand what it entails and why it is not so effective. Emotion-focused coping is discussed as not such an effective way of coping, but at times it can be an effective methods, eg. with issues that cannot be changed and need to be accepted, emotional regulation can be the most effective way of coping. Especially patient related problems cannot always be changed/addressed. 

Response: as suggested we added the definition of transference and the importance of emotion-focused coping as a possible effective coping skills [page 9]

The stress related to communication with faculty and clinical staff need to better discussed and also addressed in implications for nursing education - it seems to be a problem that needs attention from nursing education - one wonders if the faculty members are not approachable, or if the students lack communication skills - problems that can be surely be addressed. 

Response: additional discussion was provided here as well as implications for nursing education [page 8].

Reviewer 2 Report

Comments and Suggestions for Authors

Thank you for the work. It is a refreshing review to conclude the evidence for multiple systematic reviews.

A few recommendations for your consideration:

1) while the introduction provided information on the reason for the review, the rationale and justification for doing so needed to be more convincing. For instance, what are this review's implications in terms of clinical or practice or students?

2) You may have a research aim, but since the umbrella review seeks to look at findings and conclusions of existing systematic reviews and meta-analyses, you need to provide the reader with a specific area of interest or the question to be addressed.

3) In your methods, it would be good to provide some baseline information on your selection criteria for review. Also why do you only select quantitative papers? Again, it boiled down to your research questions (which was/were not indicated).

4) What is the new evidence for your findings? Need to be clearer and also to add value to the existing knowledge.

Comments on the Quality of English Language

Easily understood.

Author Response

Thank you for the constructive review. Below are my responses:

1) while the introduction provided information on the reason for the review, the rationale and justification for doing so needed to be more convincing. For instance, what are this review's implications in terms of clinical or practice or students?

Response: we added more justification and rationale of doing such umbrella review [page 2]

2) You may have a research aim, but since the umbrella review seeks to look at findings and conclusions of existing systematic reviews and meta-analyses, you need to provide the reader with a specific area of interest or the question to be addressed. 

Response: we added a research aim as suggested.[page 2]

3) In your methods, it would be good to provide some baseline information on your selection criteria for review. Also why do you only select quantitative papers? Again, it boiled down to your research questions (which was/were not indicated).

Response: we provided baseline information here and justification why only quantitative studies were included.

4) What is the new evidence for your findings? Need to be clearer and also to add value to the existing knowledge.

Response: as suggested, we provided new findings under the conclusion section. [page 10]

Reviewer 3 Report

Comments and Suggestions for Authors

Dear author

First of all, I would like to thank you for sending the manuscript. I found the research topic interesting, relevant to the journal and with a focus (umbrella review) somewhat different from other similar publications.

However, there are some aspects that are important to review.

- What is your definition of stress? Do all articles use the same definition?

- Do you always consider stress as something negative? Stress during learning can be significant for acquiring nursing skills. Students will have to face stressful situations and during the internship they must take on this challenge and face solutions for their future.

- The introduction can be improved. The author must include a framework of approach, or the importance of a new review on this topic. The introduction focuses mainly on describing the umbrella review methodology but barely addresses the topic of the study. It does not include theoretical approaches or nursing models that can serve as support.

- The information about the umbrella review design appears repeated in the introduction and methodology and discussion. Please review the wording so that the information is not repetitive.

- About the search strategy. This point is critical to understanding the results. Please detail exactly the search strings because it is not clear if there is a single search or several with the terms you indicate. Did you search with the MeSH term? Justify why you didn't do it and take it into account for the discussion.

- About the reviewers. In the data extraction and synthesis section, it states that they are carried out by two reviewers. However, there is only one author of the manuscript, and in the previous stages it seems that only one person does it. It is important for this type of study to inform who participates in designing the search, in selecting the articles to include in the sample, and who performs the analysis of the documents in the sample. The description should include how many people, whether they have experience in this type of study, what criteria they followed, or, in case of disagreement, how they resolved differences of opinion. These characteristics are important for the study and, furthermore, essential for Table 1, which appears to have scores from a single researcher.

- Table 2 may be the first of the results because it derives from the researchers' analysis.

- The themes and categories established within the themes must be defined. This information can appear in the methodology. Example: What do you understand by "coping skills"? Define each subtype. This information is generally common, but there may be differences between countries.

- In relation to the differences between countries, did you take into account the differences in training between countries? Not all countries have the same years of university studies. Not all countries require the same from clinical practices. I understand that this information corresponds to the original research (to the studies it includes) but it must present a critical analysis in this sense to improve quality and show that it is aware that not all countries provide the same nursing training. In some countries, advanced technical skills are required, you are always accompanied by an instructor, etc.

- Discussion. The discussion hardly provides new references and is mainly a summary of the results. The discussion should try to find similar studies with which to contrast. Additionally, nothing is included about the limitations of the study. The author must be critical of his work and put it in context with other similar studies.

- Relevance for Nursing education. If the author includes a critical analysis of the differences between countries, the information in this section can be improved with a more in-depth analysis.

I would like to encourage you to review these aspects to improve your manuscript because in general it is interesting and methodologically correct, but you must start from a more robust framework and provide some more information to be clear and replicable for potential readers.

Author Response

Below are my responses.

What is your definition of stress? Do all articles use the same definition?

Response: we added a definition of stress under the introduction section [page1]

Do you always consider stress as something negative? Stress during learning can be significant for acquiring nursing skills. Students will have to face stressful situations and during the internship they must take on this challenge and face solutions for their future.

Response: we added a few sentences regarding the benefit that can be derived from a manageable stress levels. [page 1]

The introduction can be improved. The author must include a framework of approach, or the importance of a new review on this topic. The introduction focuses mainly on describing the umbrella review methodology but barely addresses the topic of the study. It does not include theoretical approaches or nursing models that can serve as support.

Response: as suggested, we added a theoretical model in here.

The information about the umbrella review design appears repeated in the introduction and methodology and discussion. Please review the wording so that the information is not repetitive.

Response: we revisited this section and revised accordingly.

About the search strategy. This point is critical to understanding the results. Please detail exactly the search strings because it is not clear if there is a single search or several with the terms you indicate. Did you search with the MeSH term? Justify why you didn't do it and take it into account for the discussion.

- About the reviewers. In the data extraction and synthesis section, it states that they are carried out by two reviewers. However, there is only one author of the manuscript, and in the previous stages it seems that only one person does it. It is important for this type of study to inform who participates in designing the search, in selecting the articles to include in the sample, and who performs the analysis of the documents in the sample. The description should include how many people, whether they have experience in this type of study, what criteria they followed, or, in case of disagreement, how they resolved differences of opinion. These characteristics are important for the study and, furthermore, essential for Table 1, which appears to have scores from a single researcher.

Response: Two researchers extracted and synthesize the data. The primary researcher, who was also the author of the study, worked alongside a second independent researcher not affiliated with the study to enhance objectivity and reliability in the process. Table 1 was the scores gathered from 2 researchers. [page 5]

- Table 2 may be the first of the results because it derives from the researchers' analysis.

Response: we rearrange the table sequence

- The themes and categories established within the themes must be defined. This information can appear in the methodology. Example: What do you understand by "coping skills"? Define each subtype. This information is generally common, but there may be differences between countries.

Response: as suggested we added definition of the themes

- In relation to the differences between countries, did you take into account the differences in training between countries? Not all countries have the same years of university studies. Not all countries require the same from clinical practices. I understand that this information corresponds to the original research (to the studies it includes) but it must present a critical analysis in this sense to improve quality and show that it is aware that not all countries provide the same nursing training. In some countries, advanced technical skills are required, you are always accompanied by an instructor, etc.

Response: we provided analysis on the possible differences accross countries. In our review, due to limited contextual analysis by the reviews included, it was challenging to compare differences across countries that might affect stress and coping in students becomes challenging.

- Discussion. The discussion hardly provides new references and is mainly a summary of the results. The discussion should try to find similar studies with which to contrast. Additionally, nothing is included about the limitations of the study. The author must be critical of his work and put it in context with other similar studies.

Response: we added a section on limitations of the study. We also added similar studies and compare it with the findings of our review.

- Relevance for Nursing education. If the author includes a critical analysis of the differences between countries, the information in this section can be improved with a more in-depth analysis.

Response: we further enriched this section as suggested.

Reviewer 4 Report

Comments and Suggestions for Authors

The article presents a synthesis of the evidence focus on stress during the education of nurses in clinical training. It highlights that nurse educators play a crucial role in developing stress reduction measures and strengthening nursing students' coping skills.

Provides sufficient background and include all relevant references to the research, presents have an appropriated design based on JBI methodology. However, don’t demonstrates de quality evaluation of the articles include (sample). Discussion supported by recent evidence. Good conclusions

Comments on the Quality of English Language

Minor editing of English language required

Author Response

Below are my responses:

Provides sufficient background and include all relevant references to the research, presents have an appropriated design based on JBI methodology. However, don’t demonstrates de quality evaluation of the articles include (sample). Discussion supported by recent evidence. Good conclusions

Response: we provided sufficient details on the background as suggested.

Round 2

Reviewer 2 Report

Comments and Suggestions for Authors

Thank you for the revision. I have no further comments.

Author Response

No additional comments were made by reviewer 2.